# Metabolomics of Plasma in XLH Patients with Arterial Hypertension: New Insights into the Underlying Mechanisms

**DOI:** 10.3390/ijms25063545

**Published:** 2024-03-21

**Authors:** Luis Carlos López-Romero, José Jesús Broseta, Marta Roca-Marugán, Juan R. Muñoz-Castañeda, Agustín Lahoz, Julio Hernández-Jaras

**Affiliations:** 1Department of Nephrology, Consorci Hospital General Universitari de València, 46014 Valencia, Spain; 2Department of Nephrology and Renal Transplantation, Hospital Clínic of Barcelona, 08036 Barcelona, Spain; jjbroseta@clinic.cat; 3Metabolomics Unit, Health Research Institute Hospital La Fe (IIS La Fe), 46026 Valencia, Spain; marta_roca@iislafe.es; 4Maimonides Institute for Biomedical Research of Córdoba (IMIBIC), Nephrology Clinical Management Unit, Reina Sofia Hospital/University of Cordoba, 14071 Córdoba, Spain; juanr.munoz.exts@juntadeandalucia.es; 5Biomarkers and Precision Medicine Unit, Health Research Institute-Hospital La Fe, 46026 Valencia, Spain; agustin.lahoz@uv.es; 6Department of Nephrology, Hospital Universitari i Politècnic La Fe, 46026 Valencia, Spain; hernandez_jul@gva.es

**Keywords:** X-linked hypophosphatemia, XLH, hypertension, cardiovascular risk, fibroblast growth factor 23, FGF23, tubulopathy, hypophosphatemia

## Abstract

X-linked hypophosphatemia (XLH) is a rare genetic disorder that increases fibroblast growth factor 23 (FGF23). XLH patients have an elevated risk of early-onset hypertension. The precise factors contributing to hypertension in XLH patients have yet to be identified. A multicenter cross-sectional study of adult patients diagnosed with XLH. Metabolomic analysis was performed using ultra-performance liquid chromatography (UPLC) coupled to a high-resolution mass spectrometer. Twenty subjects were included, of which nine (45%) had hypertension. The median age was 44 years. Out of the total, seven (35%) subjects had a family history of hypertension. No statistically significant differences were found between both groups for nephrocalcinosis or hyperparathyroidism. Those with hypertension exhibited significantly higher levels of creatinine (1.08 ± 0.31 mg/dL vs. 0.78 ± 0.19 mg/dL; *p* = 0.01) and LDL-C (133.33 ± 21.92 mg/dL vs. 107.27 ± 20.12 mg/dL, *p* = 0.01). A total of 106 metabolites were identified. Acetylcarnitine (*p* = 0.03), pyruvate *p* = (0.04), ethanolamine (*p* = 0.03), and butyric acid (*p* = 0.001) were significantly different between both groups. This study is the first to examine the metabolomics of hypertension in patients with XLH. We have identified significant changes in specific metabolites that shed new light on the potential mechanisms of hypertension in XLH patients. These findings could lead to new studies identifying associated biomarkers and developing new diagnostic approaches for XLH patients.

## 1. Introduction

X-linked hypophosphatemia (XLH) is a genetic disorder caused by mutations that inactivate the X-linked phosphate-regulating endopeptidase homolog (PHEX) gene. This condition increases fibroblast growth factor 23 (FGF23), a hormone that regulates the sodium-dependent phosphate cotransporters NPT2a and NPT2c in the renal proximal tubules. As a result, XLH patients experience hypophosphaturia, hypophosphatemia, and reduced synthesis of active vitamin D (1.25(OH)_2_ vitamin D) [1]. XLH affects 1 in every 20,000 individuals, classifying it as a rare disease according to European criteria [2]. XLH exhibits a broad spectrum of symptoms, spanning from isolated hypophosphatemia to severe manifestations, including growth retardation, rickets, lower extremity deformities, bone and muscle pain, osteoarthritis, enthesopathy, and an elevated risk of bone fractures. The clinical manifestations vary between the pediatric and adult populations and among individuals within the same family [3]. 

Although controversial, some studies indicate a potential link between XLH patients and an elevated risk of early-onset hypertension, along with a greater prevalence of hypertension compared to the general population. In XLH patients, it has been suggested that hypertension may be associated with the presence of secondary or tertiary hyperparathyroidism, nephrocalcinosis, and/or impaired renal function [4,5]. Furthermore, there is evidence of the influence of FGF23 on renal sodium reabsorption and its effect on vessels and cardiomyocytes, which could potentially contribute to the presence of hypertension and left ventricular hypertrophy in this population [1,6]. In addition, there is evidence in observational studies that suggests that vitamin D level influences blood pressure and the risk of hypertension in patients with chronic kidney disease and normal subjects. However, these data have not been confirmed in randomized controlled trials [7].

Hypertension is a significant public health burden and one of the main cardiovascular risk factors and causes of mortality in the general population [8]. As we know, hypertension is a complex condition characterized by the involvement of multiple physiological pathways and organ systems. Metabolomics is one of the “omics” sciences that quantifies the changes in multiple metabolites at once and provides a more precise and complete image of an organism’s molecular structure and function. Several studies have successfully applied metabolomic techniques to characterize hypertension and suggest possible etiological pathways involving lipid, amino acid, and alcohol metabolism. As metabolites are closely linked to phenotypes, studying metabolomics can be an effective way to better understand the biology of hypertension [9].

Despite all this, it remains to be established whether the factors that could cause hypertension in this disorder are mainly due to XLH, secondary to its complications, the treatment, or other proteins and metabolites involved in phosphate homeostasis. For all this, metabolomics could be helpful to explore the associated metabolic alterations and better characterize the pathophysiology of hypertension in patients with XLH. In this study, we performed a systematic metabolomic analysis in XLH patients with and without hypertension. Our study aimed to explore the metabolomic signatures of hypertension in patients with XLH, identifying metabolites that can provide information on the pathophysiology of this disease and future lines of research.

## 2. Results

### 2.1. Characteristics of the Study Participants

The baseline characteristics of twenty adult patients diagnosed with XLH are outlined in Table 1. The study included 9 (45%) XLH patients with high blood pressure and 11 (55%) XLH patients with normal blood pressure. The gender distribution was equal, with 50% men and 50% women. Among the hypertensive group, six (66%) were men, compared to four (40%) in the normotensive group. There were no statistically significant differences between genders. 

The median age of all individuals was 44 (33.2–54.7) years. The median weight, height, and body mass index (BMI) were 73 (50.7–85.5) kg, 158 (150–170.5) cm, and 27.5 (22–32.2) Kg/m^2^, respectively. The diagnosis of XLH was made in childhood for ten subjects (50%), with five in each group. The median age at hypertension diagnosis was 35 years. Out of the total, seven (35%) subjects had a family history of hypertension, three (33.3%) in the hypertensive group and four (40%) in the non-hypertensive group. Nephrocalcinosis was present in three (15%) subjects, two (22.2%) of whom had hypertension, and one (10%) did not. In the group with hypertension, four (44.4%) had hyperparathyroidism compared to three (30%) in the normal-blood-pressure group, resulting in a total of seven (35%) patients. Nonsteroidal anti-inflammatory drugs were taken by four (20%) subjects, of which three (30%) did not have hypertension, and only one (11.1%) did. No statistically significant differences were found between hypertensive and normotensive subjects for all these variables.

Out of the total individuals, the mean SBP was 124.75 ± 17.66 mmHg, 139.77 ± 10.67 in XLH subjects with hypertension, and 112.45 ± 11.50 mmHg in normotensives, showing a statistically significant difference (*p* = 0.001). Moreover, the mean DBP for the group was 71.5 ± 7.72 mmHg, 75.66 ± 8.93 in the hypertensive group, and 68.09 ± 4.59 mmHg in normotensives (*p* = 0.04).

Out of the subjects with hypertension, six (66.6%) were taking one antihypertensive drug, two (22.2%) were taking two drugs, and one (11.1%) was taking three drugs. Regarding antihypertensive treatment, seven (77.7%) hypertensive patients were on ACE inhibitors/ARBs, five (55.5%) on calcium channel blockers, one (11.1%) on beta-blockers, and one (11.1%) on diuretics. 

### 2.2. Biochemical Analyses 

Compared to XLH subjects with normal blood pressure, those with hypertension exhibited significantly higher levels of creatinine (1.08 ± 0.31 mg/dL vs. 0.78 ± 0.19 mg/dL; *p* = 0.01) and LDL-C (133.33 ± 21.92 mg/dL vs. 107.27 ± 20.12 mg/dL, *p* = 0.01) and a lower estimated glomerular filtration rate (79.77 ± 19.75 mL/min/m^2^ vs. 103.45 ± 20.51 mL/min/m^2^, *p* = 0.01). There were no statistically significant differences found between the two groups for uric acid, total cholesterol, HDL-C, triglycerides, hemoglobin, alkaline phosphatase, phosphorus, calcium, intact FGF23, 1.25(OH)_2_D, iPTH, or TmP/GFR as observed in Table 1.

### 2.3. Metabolomic Results 

A comprehensive list of 106 identified metabolites is presented in Table 2. The table provides detailed information on the identified metabolites, including their chemical composition and properties. The Volcano graph showed that acetylcarnitine, pyruvate, ethanolamine, and butyric acid significantly differ between the blood samples of XLH patients with hypertension and those of XLH non-hypertensive patients (Figure 1). Box and whisker plots were used to represent the relative intensities of selected metabolic variables with possible clinical relevance (Figure 2). PCA score plots (Figure 3) were constructed for all metabolites and the discriminative ones, showing that the model discriminates between the two groups. From this model, the most important discriminant variables were selected according to their VIP score (>1), a jack-knife confidence interval that did not include zero, and an FC greater than 1.2. There was no correlation observed between acetylcarnitine, pyruvate, butyric acid, or ethanolamine and the levels of PTH, vitamin D, phosphorus, or FGF23 (Appendix A).

## 3. Discussion

Hypertension is a chronic condition and a significant risk factor for cardiovascular diseases (CVD). It is estimated that around 20% of adults worldwide are affected by hypertension, of which 90% of cases are classified as “essential” due to an unknown underlying cause [10]. This percentage is higher in XLH patients, who, despite being young, are at a higher risk of morbidity, mortality, and cardiovascular diseases. The prevalence of hypertension in our study comprises 45%, a figure similar to other previously described studies [4]. However, this is controversial, as other cohorts do not find similar results, and the exact pathophysiological mechanisms involved are still unclear.

The development of hypertension is influenced by a combination of genetic and environmental factors, with age, obesity, and physical inactivity being typical risk factors, and is strongly associated with inflammation, oxidative stress, and dyslipidemia [11]. Not in vain, recent studies have shown that alterations in lipid and carbohydrate metabolism often accompany hypertension [12,13], highlighting the connection between hypertension and metabolic disorders. Thus, researchers have increasingly utilized metabolomic profiles to identify specific metabolites that could serve as biomarkers of oxidative stress and inflammation, conditions commonly present in hypertensive patients [14]. With this same idea, our study, employing an untargeted metabolomic technique to identify differences in plasma metabolites between XLH patients with and without hypertension, identified a total of 106 metabolites, among which 4 showed significant clinical and statistical differences. 

Firstly, we found higher levels of acetylcarnitine, a metabolite that plays a key role in lipid metabolism, in hypertensive XLH subjects compared to normotensive individuals. Endogenous carnitine biosynthesis occurs in the kidney, liver, and brain and comes from protein lysosomal degradation [15]. The human body produces more than 1200 fatty acids (FAs) using L-carnitine and acylcarnitines [16]. The essential role of L-carnitine and acylcarnitines is to facilitate the transportation of long-chain fatty acids (long-chain fatty acyl-CoA) to the mitochondria for β-oxidation degradation. In contrast, medium and short-chain fatty acids can enter the mitochondria freely through diffusion. This function is crucial to prevent the accumulation of long-chain fatty acids that can cause cellular damage [17]. 

The analysis of serum acylcarnitine levels serves as a measurable indicator of β-oxidation status, consistently tied to vascular inflammation in various research outcomes [18]. The buildup of acylcarnitine intermediates in extracellular fluid indicates compromised β-oxidation and disrupted mitochondrial metabolism, especially in aging, leading to inefficient fatty acid recycling. On the other hand, it has been suggested that an accumulation of intracellular lipids and acetylcarnitine can occur in hypertensive subjects if the absorption rate of fatty acids exceeds the rate of β-oxidation [19]. Additionally, acylcarnitines constitute some of the most commonly implicated metabolites in cardiovascular diseases [20]. Recent clinical studies have linked acylcarnitines to an elevated risk of atherosclerotic plaque formation, independent of conventional cardiovascular risk factors [21]. Notably, Ramipril was found to significantly reduce the levels of propionylcarnitine, butyrylcarnitine, and isovalerylcarnitine (C3M, C4M, and C5M) in rats among 60 metabolites analyzed that were not affected by treatment with ACE inhibitors [22]. Hence, their role as cardiovascular risk markers in this type of population remains to be studied. 

Secondly, we observed lower levels of pyruvate in hypertensive subjects with XLH. There are limited data directly associating pyruvate levels with hypertension, but this correlation was positively made in a study involving restricted participants [19]. Conversely, one study highlighted that decreased alanine and pyruvate levels in people with hypertension may indicate a dynamic balance between aminotransferase reactions in the muscle and the liver [23,24]. While some authors propose that hypertensive patients have an elevated lactate-to-pyruvate (L/P) ratio compared to normotensive individuals [25], this ratio would reflect their response to oxygen and glucose supply variations, suggesting its role as a potential marker for tissue ischemia in anaerobic conditions. The higher L/P ratio might indicate the occurrence of such conditions in hypertensive tissues. Nevertheless, our study did not detect changes in the L/P ratio or a decrease in alanine levels.

Thirdly, elevated levels of ethanolamine were found in hypertensive patients with XLH as compared to the control group. This observation is in line with a study that involved hypertensive pediatric subjects, where ethanolamine proved to be effective in identifying children with hypertension [19]. While only a few studies have focused on the role of this metabolite in arterial hypertension, existing evidence highlights its crucial involvement in the synthesis of phosphatidylethanolamine, glycerophospholipids linked to the formation of atherosclerotic lesions and endothelial damage [26]. Furthermore, its impact on left ventricular function and the development of cardiac fibrosis has been substantiated [27].

Finally, the levels of butyric acid were also significantly higher in the hypertensive XLH population compared to normotensive individuals. Traditionally, cardiovascular research focused mainly on the involvement of long-chain fatty acids. However, increasing evidence indicates a significant impact of molecules produced by the gut microbiota, such as short-chain fatty acids (SCFA), on atherosclerosis and hypertension [28]. Butyric acid (BA) is a significant SCFA that acts as a mediator between the intestinal microbiota and the circulatory system [29]. The role of butyric acid in blood pressure BP was elucidated more than 50 years ago by intravenous administration of tributyrin, a BA prodrug [30]. Another study indicates that an elevation in the concentration of BA in the colon induces a notable hypotensive effect regulated by signals from the afferent vagus nerve of the colon. This bacterial metabolite is believed to exert its hypotensive effect by inhibiting sympathetic activity and inducing a direct vasodilatory effect [31]. In addition, BA suppresses the activation of NF-κB and reduces macrophage migration, thus assuming an anti-inflammatory role and improving atherosclerosis. In light of these findings, it is proposed that SCFAs, particularly butyric acid, could serve as a therapeutic target for atherosclerosis by modulating lipid metabolism and mitigating inflammation [32].

It is important to acknowledge certain limitations of this study. Firstly, the study’s design prevents us from establishing a causal relationship. Secondly, the small sample size necessitates categorizing this study as exploratory and hypothesis-generating until further validation is performed with larger cohorts. It also makes a potentially high type II error. In fact, our study found no correlation between hypertension and well-known related factors with high BP in the general population, such as hyperparathyroidism, nephrocalcinosis, high FGF23 levels, obesity, or the use of nonsteroidal anti-inflammatory drugs, which XLH patients commonly use for pain control. Nonetheless, it did afford us the statistical power required to detect these observed metabolite differences. Thirdly, we did not include unaffected controls with and without hypertension. Fourthly, it is important to consider the effect of medications on blood pressure and metabolism in our analysis. In our study, most patients take minimal medication in addition to active vitamin D and oral phosphorus supplements, but, maybe due to limited statistical power, the study has not yielded significant results on this point. Likewise, the study was conducted under similar conditions, with all patients fasting. The impact of metabolic regulation on flora has not been explored in this manuscript. 

Despite the outlined limitations, it is crucial to recognize the substantial strengths present in this study. Our XLH patient cohort is one of the largest published from Europe and the largest among adult XLH patients in Western Europe. Therefore, despite the potential insights gleaned from larger populations, every scientific contribution is significant in this rare disease. The field of omics sciences, such as proteomics, genomics, and metabolomics, is undergoing a surge in medical research, and we are confident that it will eventually provide us with extensive data to understand the pathophysiological mechanisms of diseases, particularly hypertension. Manipulating metabolic pathways and achieving personalized medicine represent the future of medicine.

In conclusion, the changes observed in the metabolite profile suggest that metabolomics can be a useful tool for better understanding the pathogenic mechanisms of hypertension in XLH. This can lead to earlier diagnosis, better treatment options, and the prevention of cardiovascular complications. Our study, which represents the first on hypertension in patients with XLH using a metabolomic approach, identified significant changes in specific metabolites. These findings can help us better understand this disease’s biological and pathological processes and its regulatory mechanisms, which are still unclear.

## 4. Materials and Methods

### 4.1. Study Design and Participants

This multicenter cross-sectional study was conducted on a population of adult patients diagnosed with XLH and under follow-up at the Hospital Universitari i Politècnic La Fe and other areas of the Valencian Community. The systematic identification of this cohort was performed using electronic medical records, following a methodology previously published by our group [33]. The population was stratified into two groups based on the presence or absence of hypertension. Demographic, clinical, and laboratory variables, as well as treatments used, were collected. Clinical data and treatments were collected through electronic health records, which were later validated and extended through medical interviews. The study was conducted according to the guidelines of the Declaration of Helsinki and approved by the Institutional Review Board and Ethics Committee of La Fe Health Research Institute (protocol code 2021-016-1, date of approval 27 January 2021). Informed consent was obtained from all subjects involved in the study.

### 4.2. Measurement of Blood Pressure and Covariables 

Body weight and height were measured twice, and the mean values were used to estimate BMI, calculated as weight in kilograms divided by height in square meters. Hypertension was defined as systolic blood pressure (BP) ≥ 130 mm Hg and/or diastolic BP ≥ 80 mm Hg according to the 2017 ACC/AHA guidelines [10]. In all cases of hypertension, the diagnosis had already been made by a primary care physician or another specialist prior to the commencement of our study. BP measurements were obtained during medical visits, with the primary objective of validating the diagnosis and assessing its management status. It was measured following standard protocols and by trained nurses or physicians using validated devices. The participants were in a relaxed, sitting position for 5 min and avoided caffeine, cigarettes, alcohol, and physical activity for 12 h before the clinic visit. At the medical examination, three BP measurements were recorded at 5 min intervals, and the mean BP was calculated based on the last two values. Patients diagnosed with secondary arterial hypertension were excluded. Nephrocalcinosis was defined as diffused calcium deposition within the kidney, identifiable through ultrasonography, whereas renal dysfunction was determined according to KDIGO criteria, eGFR < 60 mL/min/1.73 m^2^, and/or the presence of structural damage [34]. Hyperparathyroidism (HPT) was diagnosed in cases exhibiting consistently elevated intact parathormone (PTH) levels (>65 pg/mL) for six months or more.

### 4.3. Blood Sample Collection and Preparation

All patients included in the study were scheduled for a single medical visit at the research unit within the Nephrology Service of Hospital Universitari i Politècnic La Fe. A comprehensive evaluation was carried out during this visit, including blood tests, urine sampling, and kidney ultrasounds. All patients observed a minimum 8 h fasting period before undergoing these assessments. Blood samples were collected using ethylenediaminetetraacetic acid (EDTA) tubes and processed within 30 min to prevent platelet activation, protein degradation, and the decomposition of thermolabile compounds. Intact FGF23 levels were determined using an enzyme-linked immunosorbent assay (ELISA) test kit from Kainos Laboratories, Tokyo, Japan. Total cholesterol, high-density lipoprotein cholesterol (HDL-C), low-density lipoprotein cholesterol (LDL-C), and triglycerides were assayed using enzymatic procedures. The CKD-EPI formula was used to calculate the estimated glomerular filtration rate (eGFR), while the remaining biochemical variables were examined following the established protocols and best-practice guidelines outlined by Universitari I Politècnic La Fe Hospital.

For the metabolomic study, serum samples were promptly centrifuged at room temperature within 15 min of extraction, using the Eppendorf Centrifuge 5702 model at 5000 rpm for 10 min. The resulting serum was then divided into 250 μL aliquots, each labeled with an anonymized numerical code, before being frozen at −80 °C using a New Brunswick Scientific Ultra-Low Temperature Freezer V 410 Premium, ensuring optimal conditions for further analysis.

Upon thawing, 20 µL of serum was combined with 180 µL of cold methanol, an organic solvent that dissociates the protein-bound metabolites. The samples were then shaken at −20 °C for 30 min, inducing protein precipitation, before being centrifuged at 16,000 rpm for 20 min at 4 °C to separate the cellular fraction from the plasma. Then, 90 µL of the upper cleaned was collected from each sample and transferred to an HPLC vial and combined with 10 µL of a standard solution containing internal standards (a mixture of 20 µM caffeine-d9, leucine enkephalin, reserpine, and phenylalanine-d5). 

Quality control samples (QCs) were prepared by combining 10 μL from each extract. Blank samples, prepared to replace the extract with ultrapure water, were used to identify artifacts from reagents, the tube, and other materials. Finally, samples, QCs, and blank samples were injected randomly into the chromatographic system. To monitor the stabilities of the instrumental system and the instrumental drift, QC samples were injected into every 7th sample in each sequence, and the blank samples were performed at the end of the sequence.

### 4.4. Untargeted Metabolomics Based on UPLC-Q-ToF Mass Spectrometry

The metabolomic analysis was performed using Ultra-Performance Liquid Chromatography (UPLC) equipment coupled to a high-resolution mass spectrometer (MS) with Orbitrap UPLC-QExactive Plus detector (UPLC-TOF/MS-Orbitrap, QExactive Plus MS) available at the Analytical Unit of the Instituto de Investigación Sanitaria La Fe (IISLaFe). All reagents and chemicals were purchased from Sigma Aldrich (St. Louis, MO, USA) and Fisher Scientific (CAN, Waltham, MA, USA).

Chromatographic separation was performed using an HILIC UPLC X bridge BEH amide column (150 mm × 2.1 mm, particle size 2.5 uM, Waters, Wexford, Ireland). The column and auto counter were set to 25 °C and 4 °C, respectively, with an injection volume of 5 μL. The total running time of the chromatogram was 25 min at a flow rate of 105 μL/min. Metabolites were eluted with gradients of mobile phase A consisting of H_2_O and 10 mM ammonium acetate and mobile phase B in acetonitrile (ACN). During the first 2 min, the eluent composition was 90% mobile phase B and 10% mobile phase A; at minute 2, 85% B; from minute 3 to minute 8, 75% B; from minute 8 to minute 10, 70% B; from minute 10 to minute 13, 50% B; from minute 13 to minute 16, 25% B; from minute 16 to minute 22, 0% B; from minute 22 to minute 25, again 90% B reaching the starting conditions. A mass detector (Orbitrap) was used in Full scan in both positive and negative ESI modes with two different events, the first scan event between 70 and 700 Da and a second event in the mass range of 700–1700 Da. 

Different measures were taken to ensure the analysis’s quality and reproducibility and mitigate intra-batch variability. The reagent blank was analyzed at the beginning of the sequence to identify sampling artifacts and exclude contaminating signals. Subsequently, a random injection sequence was performed that included the analysis of 5 quality control (QC) samples to condition the column and equipment (these data were not used for data analysis). Sample analyses and quality control were conducted at a 5:1 ratio to monitor and control variability resulting from contamination or changes in the column. Any signal detected with an intensity similar to the quality control and blank samples was eliminated.

### 4.5. Data Processing and Acquisition

The data obtained during the analysis were converted into mzXML format using the Mass Converter Proteowizard program. Then, they were processed in EI-MAVEN software (Version Version 0.4.1) for alignment, noise filtering, the integration of chromatographic peaks, generation of a peak table, and identification of metabolites using an in-house library of polar compounds previously developed and used at the Analytical Unit [35]. Structures, *m*/*z* spectra, and formulas were obtained from the Human Metabolome Database (HMDB, version 4.0).

The configuration parameters of EI-MAVEN for data processing were the following: ionization in automatic mode type ESI, precision q1 and q3 for MRN transitions of 0.5 amu with total filter line. Maximum differences in retention time between spikes in a group were 15 s. The smoothness threshold was 2 λ, and the asymmetry was 0.08 ρ. A weighted average of the m/z values observed in the chromatographic peak was calculated to annotate the chromatographic peaks. Cluster selection criteria for use in the alignment were at least 2 peaks in each cluster with a total limit of 1000 clusters and a clustering window of 20 scans. For peak picking, the minimum intensity was 1000 with a maximum signal-to-noise ratio of 3 and a minimum peak width of 5 scans. The peaks are assigned a clustering score.

We have applied the NetID algorithm to the dataset, creating a peak table containing the exact mass (*m*/*z*) and retention time (min). Subsequently, matches were sought with those metabolites in the database. The predefined tolerance was ±10 ppm. Peaks were considered metabolites or artifacts based on the match with the local database formulas. All adducts, fragments, or isotopes of metabolites were considered artifacts.

When comparing with blank samples, background peaks were removed if their intensity was greater than 0.5 times that of the sample. Compounds showing a coefficient of variation (CV) in the analytical response of the quality controls exceeding 30% were excluded.

### 4.6. Statistical Analysis 

Categorical variables were presented as *n* (%), while continuous variables were presented as median (P25–P75). The distribution of the variables was tested with the Shapiro–Wilk test. Differences between hypertensive and normotensive groups were analyzed using independent samples *t*-test for normally distributed continuous data, the Mann–Whitney U test for skewed data, and the χ^2^ test for categorical variables. 

The relative abundances of metabolic members of XLH individuals with hypertension were compared with non-hypertensive XLH controls. A univariate statistical analysis was conducted between the case and control groups utilizing a *t*-test and visualized through a Volcano Plot. This analysis was executed using an in-house script within the R platform (Version 4.3.3). The methodology integrated a Fold Change (FC) approach alongside the significance derived from a paired Student’s *t*-test for normally distributed variables. For skewed data, the analysis employed the Wilcoxon signed-rank test after confirming normality via the Shapiro–Wilk test.

Molecular characteristics demonstrating the stronger combination of FC and statistical significance are represented in Figure 1. To mitigate the risk of false discoveries, we applied the Benjamini–Hochberg procedure. An exploratory unsupervised Principal Component Analysis (PCA) was performed to extract maximum information and identify behavioral patterns. This involved simplifying data variability by examining their distribution, ultimately grouping them into principal components. Model validity and robustness were assessed using R2(Y) (goodness of fit) and Q2(Y) (goodness of prediction), with a predictive capacity deemed satisfactory when Q2(Y) exceeded 0.5. A PCA score plot accompanies each comparison. For univariate analysis, *p* < 0.05 was defined to reach statistical significance in a two-tailed Student’s *t*-test. Metabolic compounds with *p* < 0.05 and variable importance in the projection (VIP) > 1 were considered statistically different between groups. For statistical analysis, the SPSS system for Windows version 22 and the GraphPad Prism v8 software (San Diego, CA, USA) were used.

## Figures and Tables

**Figure 1 ijms-25-03545-f001:**
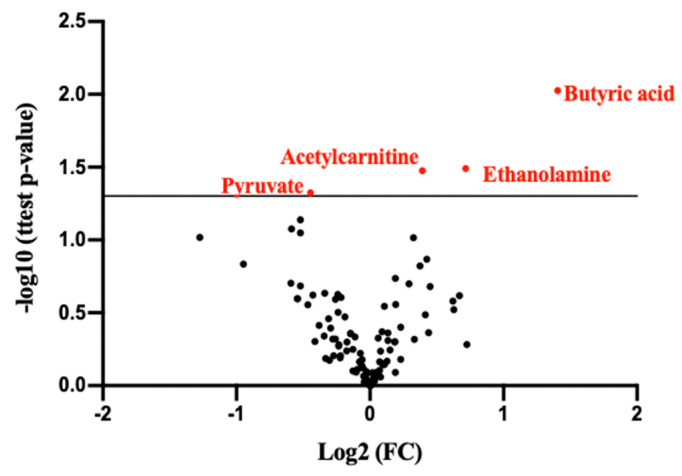
Volcano plot illustrates significant differences in serum metabolites between XLH patients with and without hypertension.

**Figure 2 ijms-25-03545-f002:**
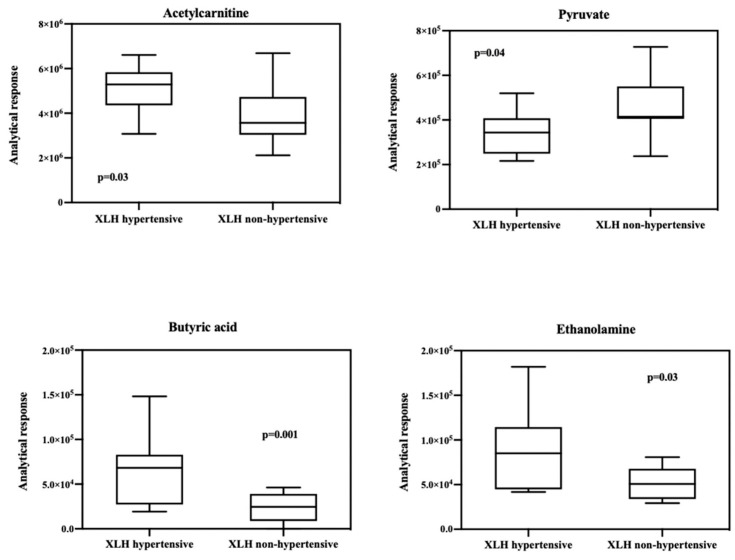
Box and whisker plot of significantly altered metabolites in the hypertensive group compared with the normotensive group.

**Figure 3 ijms-25-03545-f003:**
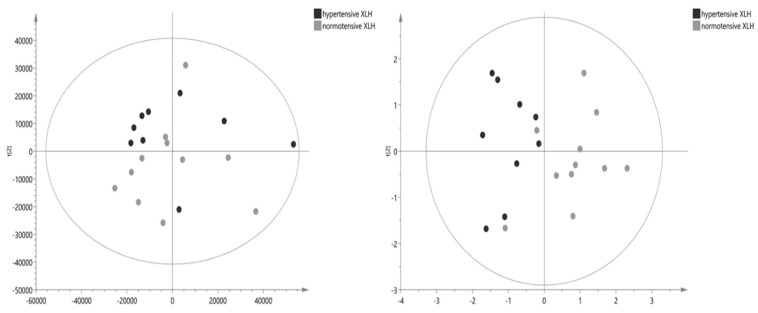
PCA score plot with metabolic variations between hypertensive XLH patients and normotensive XLH patients when all metabolites and only discriminative metabolites are identified.

**Table 1 ijms-25-03545-t001:** Demographic and clinical characteristics of the study population.

Variable	Overall (*n* = 20)	XLH Hypertensive (*n* = 9)	XLH Non-Hypertensive (*n* = 11)
Age (years)	44 (33.2–54.7)	49 (39–57)	43 (27–47)
Male sex, *n* (%)	10 (50)	6 (66)	4 (40)
Weight (kg)	73 (50.7–85.5)	81.5 (50.2–93.7)	71 (50–79.7)
Height (cm)	158 (150–170.5)	159.5 (152.2–182.7)	155.5 (150–163)
BMI (Kg/m^2^)	27.5 (22–32.2)	30 (22.5–32.7)	25 (22–32)
Diagnosis in childhood, *n* (%)	10 (50)	5 (55.5)	5 (50)
Office SBP (mmHg)	124.75 ± 17.66	139.77 ± 10.67	112.45 ± 11.50
Office DBP (mmHg)	71.5 ± 7.72	75.66 ± 8.93	68.09 ± 4.59
Familiar history of hypertension *n* (%)	7 (35)	3 (33.3)	4 (40)
Nephrocalcinosis *n* (%)	3 (15)	2 (22.2)	1 (10)
Hyperparathyroidism *n* (%)	7 (35)	4 (44.4)	3 (30)
Non-steroidal anti-inflammatories *n* (%)	4 (20)	1 (11.1)	3 (30)
Creatinine (mg/dL), mean ± SD	0.91 ± 0.29	1.08 ± 0.31	0.78 ± 0.19
GFR (mL/min/1.73 m^2^), mean ± SD	92.80 ± 23	79.77 ± 19.75	103.45 ± 20.51
Uric acid (mg/dL), mean ± SD	5.69 ± 1.52	6.45 ± 1.61	5.07 ± 1.18
Total cholesterol(mg/dL), mean ± SD	197 ± 28.36	206.88 ± 32.06	189 ± 23.39
LDL (mg/dL), mean ± SD	119 ± 24.33	133.33 ± 21.92	107.27 ± 20.12
HDL (mg/dL), mean ± SD	55.3 ± 14.91	50.22 ± 16.57	59.45 ± 12.68
Triglycerides (mg/dL), mean ± SD	115.94 ± 50.38	122.62 ± 58.63	111.09 ± 45.84
Hemoglobin (g/dL), mean ± SD	14.64 ± 1.63	14.9 ± 1.29	14.26 ± 1.91
Alkaline phosphatase (UI/L), mean ± SD	97.85 ± 43.68	112 ± 53.78	86.27 ±31.37
Phosphorus (mg/dL), mean ± SD	2.14 ± 0.49	2.08 ± 0.38	2.18 ± 0.58
Calcium (mg/dL), mean ± SD	9.46 ± 0.72	9.51 ± 0.54	9.42 ± 0.86
Intact FGF23 (pg/mL), mean ± SD	211.96 ± 248.58	288.28 ± 346.29	149.52 ± 109.28
Vitamin D (ng/mL), mean ± SD	27.97 ± 9.44	25.87 ± 6.37	29.5 ± 11.22
PTH (pg/mL), mean ± SD	74.88 ± 48.87	83.34 ± 38.80	67.96 ± 56.71
TmP/GFR (mg/dL), mean ± SD	2.12 ± 0.48	2.08 ± 0.28	2.07 ± 0.53

BMI: body mass index, SBP: systolic blood pressure, DBP: diastolic blood pressure, GFR: glomerular filtration rate, LDL: low-density lipoprotein, HDL high-density lipoprotein, FGF23: fibroblast growing factor 23, PTH, parathormone, Vit D: 1.25(OH)_2_ vitamin D, TmP/GFR: ratio of tubular maximum reabsorption of phosphate (TmP) to GFR.

**Table 2 ijms-25-03545-t002:** List of identified metabolites.

Compound	Formula	*m*/*z*	Rt	Adduction	Class
glycine	C2H5NO2	74.024834	12.671	[M−H]−	Amino acid/peptides/analogues
Hypotaurine	C2H7NO2S	108.012642	11.876	[M−H]−	Sulfinic acid and derivatives
taurine	C2H7NO3S	126.021965	9.911	[M+H]+	Organosulfonic acid and derivatives
O-Phosphoethanolamine	C2H8NO4P	142.026367	13.674	[M+H]+	Phosphate esters
pyruvate	C3H4O3	87.008896	3.546	[M−H]−	Keto acids and derivatives
Guanidoacetic acid	C3H7N3O2	118.061012	12.4	[M+H]+	Amino acid/peptides/analogues
serine	C3H7NO3	104.035446	12.858	[M−H]−	Amino acid/peptides/analogues
succinate	C4H6O4	117.019478	12.021	[M−H]−	Carboxylic acid and derivatives
4-aminobutyrate	C4H9NO2	104.070511	11.303	[M+H]+	Amino acid/peptides/analogues
D-2-Aminobutyric acid	C4H9NO2	104.070511	11.303	[M+H]+	Amino acid/peptides/analogues
dimethylglycine	C4H9NO2	102.056160	11.155	[M−H]−	Amino acid/peptides/analogues
glutamine	C5H10N2O3	147.076279	12.909	[M+H]+	Amino acid/peptides/analogues
2-Hydroxyvaleric acid	C5H10O3	117.055817	3.579	[M−H]−	Fatty and conjugated acids
methionine	C5H11NO2S	148.043945	9.137	[M−H]−	Amino acid/peptides/analogues
ornithine	C5H12N2O2	131.082657	16.205	[M−H]−	Amino acid/peptides/analogues
Phosphorylcholine	C5H14NO4P	184.073257	13.76	[M+H]+	Quaternary ammonium salts
hypoxanthine	C5H4N4O	135.031296	6.754	[M−H]−	Purines and derivatives
2-hydroxyglutarate	C5H8O5	147.030060	10.955	[M−H]−	Short chain hydroxy acids and derivatives
lysine	C6H14N2O2	145.098297	17.035	[M−H]−	Amino acid/peptides/analogues
N′-Methylnicotinamide	C7H8N2O	137.070877	13.417	[M+H]+	Pyridines and derivatives
Guaiacol	C7H8O2	123.045341	2.574	[M−H]−	Phenols
2-Hydroxyoctanoic acid	C8H16O3	159.102707	2.447	[M−H]−	Fatty and conjugated acids
2-Phenylbutyric acid	C10H12O2	163.076599	2.439	[M−H]−	Benzene and derivatives
Indole-3-acetaldehyde	C10H9NO	158.061234	2.544	[M−H]−	Indoles and derivatives
DL-Indole-3-lactic acid	C11H11NO3	204.066666	2.485	[M−H]−	Indoles and derivatives
tryptophan	C11H12N2O2	205.097092	7.689	[M+H]+	Indoles and derivatives
Undecanoic acid	C11H22O2	185.154770	2.226	[M−H]−	Fatty and conjugated acids
trans-3-Indoleacrylic acid	C11H9NO2	188.07074	3.044	[M+H]+	Indoles and derivatives
Indole-3-pyruvic acid	C11H9NO3	202.051147	2.215	[M−H]−	Indoles and derivatives
Butyric acid	C4H8O2	202.087463	2.495	[M−H]−	Indoles and derivatives
Myristic acid	C14H28O2	227.201828	2.147	[M−H]−	Fatty and conjugated acids
Linoleic acid	C18H32O2	279.233337	2.101	[M−H]−	Lineolic acids and derivatives
Stearic acid	C18H36O2	283.264160	2.082	[M−H]−	Fatty and conjugated acids
Arachidic acid	C20H40O2	311.295898	2.048	[M−H]−	Fatty and conjugated acids
Chenodeoxycholic acid	C24H40O4	393.299957	2.996	[M+H]+	Bile acids, alcohols and derivatives
Glycodeoxycholic acid	C26H43NO5	450.321655	3.096	[M+H]+	Bile acids, alcohols and derivatives
Glycocholic acid	C26H43NO6	464.302124	6.826	[M−H]−	Bile acids, alcohols and derivatives
Taurodeoxycholic acid	C26H45NO6S	498.290253	2.389	[M−H]−	Bile acids, alcohols and derivatives
Glycolic acid	C2H4O3	75.008797	8.434	[M−H]−	Hydroxyacids and derivatives
acetylphosphate	C2H5O5P	140.994659	12.582	[M+H]+	Phosphate esters
lactate	C3H6O3	89.024582	7.071	[M−H]−	Hydroxyacids and derivatives
glycerate	C3H6O4	105.019455	9.939	[M−H]−	Carbohydrates and conjugates
alanine	C3H7NO2	90.054924	12.178	[M+H]+	Amino acid/peptides/analogues
b-alanine	C3H7NO2	90.054924	12.809	[M+H]+	Amino acid/peptides/analogues
sarcosine	C3H7NO2	90.054924	12.178	[M+H]+	Amino acid/peptides/analogues
Trimethylamine N-oxide	C3H9NO	76.075676	12.071	[M+H]+	Organonitrogenic compounds
Ethanolamine	C2H7N0	106.086212	11.132	[M+H]+	Organonitrogenic compounds
acetoacetate	C4H6O3	101.024567	5.873	[M−H]−	Keto acids and derivatives
2-Ketobutyric acid	C4H6O3	101.024574	2.401	[M−H]−	Keto acids and derivatives
N-Acetylglycine	C4H7NO3	116.035461	8.303	[M−H]−	Amino acid/peptides/analogues
aspartate	C4H7NO4	134.044907	13.404	[M+H]+	Amino acid/peptides/analogues
asparagine	C4H8N2O3	133.060822	13.126	[M+H]+	Amino acid/peptides/analogues
3-hydroxylbutyrate	C4H8O3	103.040222	7.334	[M−H]−	Hydroxyacids and derivatives
(R)-3-Hydroxybutanoic acid	C4H8O3	103.040222	7.700	[M−H]−	Hydroxyacids and derivatives
2-Hydroxybutyric acid	C4H8O3	103.040237	5.802	[M−H]−	Hydroxyacids and derivatives
creatine	C4H9N3O2	130.062256	12.102	[M−H]−	Amino acid/peptides/analogues
L-Homoserine	C4H9NO3	120.065437	12.502	[M+H]+	Amino acid/peptides/analogues
threonine	C4H9NO3	120.065437	12.502	[M+H]+	Amino acid/peptides/analogues
3-Hydroxyisovaleric acid	C5H10O3	117.055817	5.689	[M−H]−	Amino acid/peptides/analogues
DL-Valine	C5H11NO2	116.071785	10.032	[M−H]−	Amino acid/peptides/analogues
betaine	C5H11NO2	116.071785	10.032	[M−H]−	Amino acid/peptides/analogues
Adonitol	C5H12O5	151.061417	8.769	[M−H]−	Carbohydrates and conjugates
Choline	C5H13NO	104.106903	12.645	[M+H]+	Organonitrogenic compounds
Uric acid	C5H4N4O3	167.021011	8.487	[M−H]−	Purines and derivatives
Pyroglutamic acid	C5H7NO3	128.035507	8.453	[M−H]−	Amino acid/peptides/analogues
2-Oxoisopentanoic acid	C5H8O3	115.040176	2.285	[M−H]−	Keto acids and derivatives
Methylsuccinic acid	C5H8O4	131.035095	10.675	[M−H]−	Fatty and conjugated acids
proline	C5H9NO2	116.070442	10.857	[M+H]+	Amino acid/peptides/analogues
N-Propionylglycine	C5H9NO3	132.065613	12.203	[M+H]+	Amino acid/peptides/analogues
Aminolevulinic acid	C5H9NO3	130.051025	12.026	[M−H]−	Amino acid/peptides/analogues
L-Hydroxyproline	C5H9NO3	130.051056	7.712	[M−H]−	Amino acid/peptides/analogues
N-Methyl-D-aspartic acid	C5H9NO4	148.060257	12.906	[M+H]+	Amino acid/peptides/analogues
glutamate	C5H9NO4	146.045898	12.746	[M−H]−	Amino acid/peptides/analogues
2-Methyl-3-ketovaleric acid	C6H10O3	129.055695	2.195	[M−H]−	Keto acids and derivatives
Adipic acid	C6H10O4	145.050690	11.644	[M−H]−	Fatty and conjugated acids
DL-Pipecolic acid	C6H11NO2	130.086411	10.533	[M+H]+	Amino acid/peptides/analogues
4-Methylvaleric acid	C6H12O2	115.076584	2.533	[M−H]−	Fatty and conjugated acids
L-Rhamnose	C6H12O5	163.061142	8.018	[M−H]−	Carbohydrates and conjugates
myo-inositol	C6H12O6	179.056076	13.264	[M−H]−	Alcohols and polyols
Fructose	C6H12O6	179.056091	10.639	[M−H]−	Carbohydrates and conjugates
Mannose	C6H12O6	179.056091	10.639	[M−H]−	Carbohydrates and conjugates
citrulline	C6H13N3O3	174.088364	13.056	[M−H]−	Amino acid/peptides/analogues
leucine	C6H13NO2	130.087402	8.878	[M−H]−	Amino acid/peptides/analogues
arginine	C6H14N4O2	173.104385	16.941	[M−H]−	Amino acid/peptides/analogues
Aconitic acid	C6H6O6	173.009048	13.093	[M−H]−	Carboxylic acid and derivatives
3-Methylglutaconic Acid	C6H8O4	143.035065	10.646	[M−H]−	Fatty and conjugated acids
citrate/isocitrate	C6H8O7	191.019623	13.506	[M−H]−	Carboxylic acid and derivatives
3-Methyl-Histidine	C7H11N3O2	168.077744	12.867	[M−H]−	Amino acid/peptides/analogues
Homocitrulline	C7H15N3O3	190.1185	13.066	[M+H]+	Amino acid/peptides/analogues
gamma-Butyrobetaine	C7H15NO2	146.117508	13.111	[M+H]+	Fatty and conjugated acids
carnitine	C7H15NO3	162.11232	12.578	[M+H]+	Quaternary amine
Homoarginine	C7H16N4O2	189.134613	17.538	[M+H]+	Amino acid/peptides/analogues
Benzoic acid	C7H6O2	121.029625	10.036	[M−H]−	Benzene and derivatives
3-Hydroxybenzoic acid	C7H6O3	137.024445	2.072	[M−H]−	Benzene and derivatives
2-6-Dihydroxybenzoic acid	C7H6O4	153.019501	1.822	[M−H]−	Benzene and derivatives
N-Acetyl-L-arginine	C8H16N4O3	217.129395	12.865	[M+H]+	Amino acid/peptides/analogues
DL-2-Aminooctanoic acid	C8H17NO2	160.133209	6.675	[M+H]+	Amino acid/peptides/analogues
Glycerophosphocholine	C8H20NO6P	258.109863	13.14	[M+H]+	Glycerophospholipids
3-Indoxyl sulfate	C8H7NO4S	212.002319	1.863	[M−H]−	Sulfuric acids and derivatives
2-Phenylpropionic acid	C9H10O2	149.060959	2.475	[M−H]−	Phenylpropanoids
phenylalanine	C9H11NO2	164.071762	8.015	[M−H]−	Amino acid/peptides/analogues
tyrosine	C9H11NO3	180.066513	9.686	[M−H]−	Amino acid/peptides/analogues
Acetylcarnitine	C9H17NO4	204.123001	10.376	[M+H]+	Fatty acid esters
4-Hydroxyphenylpyruvic acid	C9H8O4	179.035172	2.368	[M−H]−	Benzene and derivatives
3-Methylindole	C9H9N	132.080872	3.064	[M+H]+	Indoles and derivatives
Hydroxyphenylformamidoacetic acid	C9H9NO4	194.045883	6.769	[M−H]−	Phenol

*m*/*z*: mass/charge, Rt: Retention time.

## Data Availability

The datasets and materials used and/or analyzed during the current study are available from the corresponding author upon reasonable request.

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
