# Peer review of "Metabolomics of Plasma in XLH Patients with Arterial Hypertension: New Insights into the Underlying Mechanisms"

_ijms, 2024, doi:10.3390/ijms25063545_

Round 1

Reviewer 1 Report

Comments and Suggestions for Authors

In this manuscript, the authors performed metabolomic analyses on plasma from XLH patients with or without hypertension, and discuss a the metabolites which differ between the 2 groups,noting that a few of them have been implicated in processes that would impact cardiovascular disease and blood pressure. This manuscript is interesting in topic as it is unclear whether XLH itself is a risk for HTN, and if there is higher risk for HTN, what the difference is in these patients. While the data is interesting, some more detailed dissection would be useful:

1)      It is striking that there are 40+ patients in this XLH cohort with HTN, but a majority of them are not on medications. Were these patients newly diagnosed?

2)      Were metabolite levels different in patients with more severe vs more mild HTN? Could the patients with HTN be stratified and see how the levels are different? And how that correlates with FGF23 , phosphate levels?

3)      How does this metabolome in XLH with and without HTN compared to that characterized in patients with essential HTN who do not have XLH? A limitation of this study is that there is no unaffected controls with and without HTN.

Comments on the Quality of English Language

clearly written

Reviewer 2 Report

Comments and Suggestions for Authors

The authors investigated the metabolomics of plasma in XLH patients to uncover mechanisms behind their increased early-onset hypertension risk. The study involved 20 XLH adults, divided into hypertensive and normotensive groups. Using UPLC-MS, 106 metabolites were identified, with acetylcarnitine, pyruvate, ethanolamine, and butyric acid significantly differing between groups. These metabolites suggest pathways involved in hypertension among XLH patients, such as impaired fatty acid and amino acid metabolism, endothelial dysfunction, cardiac fibrosis, and the role of gut microbiota-produced butyric acid. This research highlights important metabolic changes associated with hypertension in XLH, suggesting directions for further mechanism, biomarker, and treatment research.

However, reviewers noted several limitations affecting the study's validity. The small sample size compromises statistical power, risking type II errors. The cross-sectional design limits causality inference between metabolite changes and hypertension. Variability in BP measurements, unadjusted medication effects, reliance on patient recall, and lack of comparison with healthy controls were major concerns. Additionally, not documenting diet and gut flora, significant metabolic regulators, was seen as a gap. Recommendations included increasing sample size, adopting longitudinal designs, adjusting for confounders, incorporating healthy controls, and validating findings in independent datasets to enhance study design and conclusiveness, paving the way for definitive connections between metabolic pathways and BP changes in XLH.

Round 2

Reviewer 1 Report

Comments and Suggestions for Authors

Comments are addressed - for the correlation with FGF, Phos, etc, the authors state this data but do not show it in the manuscript - would useful to actually show the analyses- wither in the paper or as supplementary information

Comments on the Quality of English Language

english is fine
